

# More accurate aeroelastic wind-turbine load simulations using detailed inflow information

Mads Mølgaard Pedersen[1], Torben Juul Larsen[1], Helge Aagaard Madsen[1], and Gunner Christian Larsen[1]

[1]Wind Energy Department, Technical University of Denmark, Frederiksborgvej 399, DK-4000 Roskilde, Denmark

*Correspondence to:* Mads M. Pedersen (mmpe@dtu.dk)

**Abstract.**

In this paper, inflow information is extracted from a measurement database and used for aeroelastic simulations to investigate if using more accurate inflow descriptions improves the accuracy of the simulated fatigue loads.

The inflow information is extracted from the nearby met masts and a blade-mounted five-hole pitot tube. The met masts provide measurements of the inflow at fixed positions some distance away, whereas the pitot tube measures the inflow while rotating with the rotor.

The met mast measures the free-inflow velocity, but the measured turbulence may evolve on its way to the turbine, pass besides the turbine, or the mast may be in the wake of the turbine. The inflow measured by the pitot tube, on the other hand, is very representative of the wind that acts on the turbine as it is measured close to the blades and includes variations within the rotor plane. This inflow is, however, affected by the presence of the turbine, and therefore an aerodynamic model is used to estimate the free-inflow velocities that would have been at the same time and position without the presence of the turbine.

The inflow information used for the simulations includes the mean wind speed and trend, the turbulence intensity, wind shear profile, atmospheric stability dependent turbulence parameters, and azimuthal variations within the rotor plane. In addition, the instantly measured wind speed is used to constrain the turbulence.

It is concluded that the period-specific turbulence intensity must be included in the aeroelastic simulations to make the range of the simulated fatigue loads representative for the range of the measured fatigue loads. Furthermore, it is found that the one-to-one correspondence between the measured and simulated fatigue loads is improved considerably by using inflow characteristics extracted from the pitot tube instead of the met-mast-based sensors as input for the simulations. Finally, the use of pitot-tube wind speed to constrain the turbulence is found to decrease the variation of the simulated loads due to different turbulence realisations (seeds), such that the need for multiple simulations is reduced.

## 1 Introduction

Aeroelastic simulations are extensively used in the development of modern wind turbines. These simulations are used to estimate the dynamic response of the wind turbine structure in both the research, the design and the certification phase; they are used to investigate new concepts, evaluate different designs, and to prove that the life-time fatigue and extreme loads are below the capability limits of the components.





Aeroelastic simulations are typically based on simplified models of the wind-turbine structure, its aerodynamic properties and the inflow conditions. Often, standard or the site-average turbulence parameters and shear profile are used for the inflow modelling. This approach makes it possible to compare the simulation results with the average load level of the measurements despite the often massive measurement scatter, which is mainly caused by different inflow conditions; see the example in Fig.

5    1 (left).

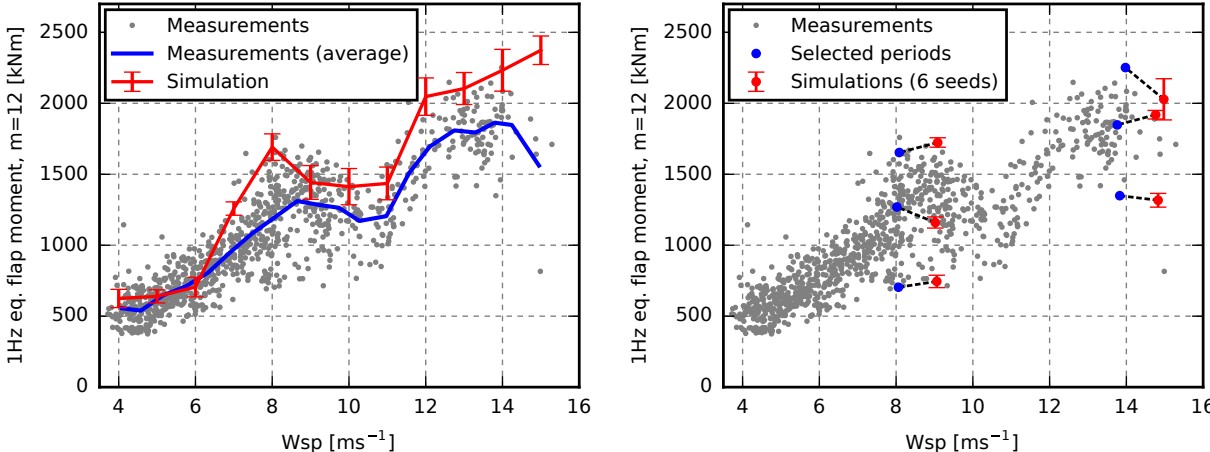

**Figure 1.** Two approaches for comparing the measured and simulated loads. Left: The traditional approach where the site-average turbulence characteristics and shear profile are used as input for the aeroelastic simulations. The results are compared to the measured average load levels. Right: The suggested one-to-one approach where measured inflow characteristics are extracted from selected time series. The simulation results are compared to the corresponding measurement observation. Note that the simulation error bars are offset 1 ms$^{-1}$ to the right to increase clarity.

In this paper, the effects of using more specific inflow characteristics for aeroelastic simulations are investigated. The idea is to select single measurement time series and extract information for more accurate inflow fields, i.e. descriptions of the mean inflow velocities, as well as the turbulent fluctuations. These inflow fields are subsequently used as input for numerical load simulations, and the simulated loads are compared to the original measured loads; see Fig. 1 (right).

10    As seen in Fig. 2, the measured blade-root fatigue load increases with the wind speed. The scatter is, however, massive. Different levels of turbulence intensity can explain some of the variation, especially for low wind speeds, but a lot of the variation is caused by a combination of other factors, e.g. variability in wind shear profile, atmospheric stability, etc. The hope is, therefore, that it will be possible to select a period of interest, extract the inflow characteristics, and thus reproduce the period in an aeroelastic simulation giving similar loads. In this way, the reason for the high loads can be investigated and

15    subsequently used to predict future loads with higher accuracy. In addition, the measurement period required for load validation can potentially be reduced by using a reduced set of single time series instead of the average of a large measurement dataset.





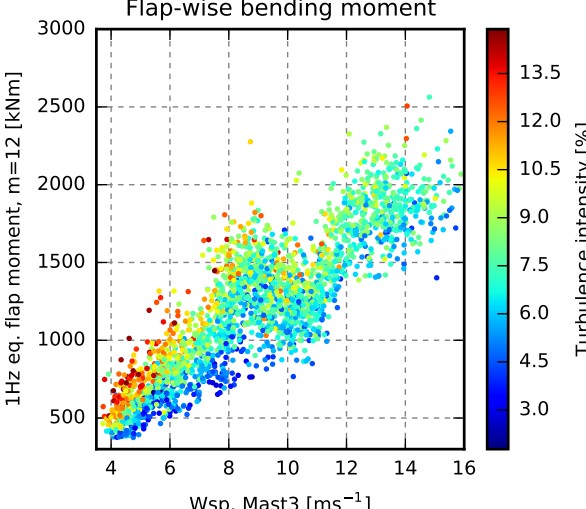

**Figure 2.** The blade-root flap-wise fatigue load plotted as a function of wind speed and coloured by turbulence intensity. The turbulence intensity affects the blade-root fatigue loads. Much of the scatter is, however, caused by other factors.

The inflow characteristics required for the description of the more accurate inflow fields can be extracted from cup or sonic anemometers at a nearby met mast, if the anemometers are exposed to similar inflow conditions. This means that the mast must be close to the turbine, but out of the rotor induction zone. Furthermore, wind directions where the anemometers are in the wake of turbines or the mast itself must be discarded, as well as situations where the turbine is in the wake of other turbines.

5 In addition, anemometers at different heights are required to measure the shear profile.

The inflow parameters can, alternatively, be obtained from a blade-mounted flow sensor, BMFS. Mounted at the blade, a BMFS is exposed to exactly the same inflow conditions as the turbine, and that goes for all wind directions. In addition, a BMFS also provides valuable information about the variation within the rotor area.

A BMFS is, however, located inside the rotor induction zone, and therefore a method to compensate for the presence of the 10 turbine is required; i.e. a method that takes the flow velocities measured relative to the BMFS and calculates the free-stream inflow velocities that would have been observed at the same time and location without the presence of the wind turbine. In this study, the method presented by Pedersen et al. (2018) is used. This method uses a combination of aerodynamic models to estimate the disturbance that the turbine induces on the free-stream inflow.

In the right setup, lidars are able to provide information similar to a BMFS – in fact, the BMFS could be a lidar as in the 15 experiment by Pedersen et al. (2013). Typically, however, lidars are mounted on the nacelle or on the ground, and set up to measure the inflow some distance up- or downstream, in which case they have other advantages and drawbacks.

The current study is based on the measurement database established during the DAN-AERO project (Madsen et al., 2010b), where a 3.6MW Siemens wind turbine at Høvsøre Test Centre for Large Wind Turbines was equipped with a blade-mounted



five-hole pitot tube. The aeroelastic simulations in this study are performed using HAWC2, which is a non-linear aeroelastic code intended for computing wind turbine response in time domain (Larsen and Hansen, 2007).

## 2   Method

From the measurement database (see Section 2.1), 20 different 10-min periods, denoted P1 - P20, are extracted. These periods

are selected to be no-wake situations representing a wide range of load levels at 8 and 14 $\mathrm{ms}^{-1}$; i.e. below and above the rated wind speed. From each period, inflow characteristics are extracted for the simulation cases, Cases 1 - 5, which utilise different details about the inflow, e.g. wind speed trend, turbulence intensity, shear etc.

For each of the selected periods, these inflow characteristics are used as input for a set of six simulations with different turbulence realisations (seeds). Finally, the simulated loads are compared to their measured counterparts.

### 2.1   Site, turbine, sensor and data overview

The measurement database used in this study was recorded from April to July 2009 as part of the DAN-AERO project (Madsen et al., 2010b; Troldborg et al., 2013). It contains 9600 data files with 10-minute measurements from a Siemens 3.6 MW wind turbine located at Høvsøre Test Site for Large Wind Turbines in Denmark, as well as measurements from the nearby met masts; see Fig. 3. The rotor diameter is 107 m and the hub height is 89.5 m. The turbine was equipped with blade-root

bending-moment sensors and a blade-mounted five-hole pitot tube.

As seen in Fig. 3, the turbine was located in the middle of a row of five megawatt wind turbines. Mast3, which is located around 2.5 diameters west of the turbine, provides hub-height wind-speed observations, while the main met mast, 820 m south of the turbine, measures the wind speed at six different heights ranging from 10 to 116.5 m.

### 2.2   Blade-mounted five-hole pitot tube

During the measurement period, an Aeroprobe CPSPY5 five-hole pitot tube was mounted on one of the blades in radius 36 m, i.e. around one third from the tip. A five-hole pitot tube measures the relative flow speed as well as the flow angle at two perpendicular planes. From this information, the relative 3D flow velocity can be calculated, and subtracting the velocity due to sensor movement yields the flow velocity in the rotor plane; see Pedersen et al. (2017) for more details. In this study, the velocity due to sensor movement is calculated based on the rotor rotation and the pitch motion. This means that movement due

to dynamic tower and blade deflection is not included, and some discrepancy is therefore expected.

The flow velocity is mapped from the rotating blade section coordinate system to fixed global coordinates. In this process, additional uncertainty is introduced, as the exact orientation of the blade section is unknown due to the deflection and torsion of the structure.

Finally, the wind turbine induction, i.e. the disturbance of the inflow field caused by the presence of the rotor, is estimated us-

ing a combination of aerodynamic models. In this study, the aerodynamic models comprise blade-element-momentum (BEM)



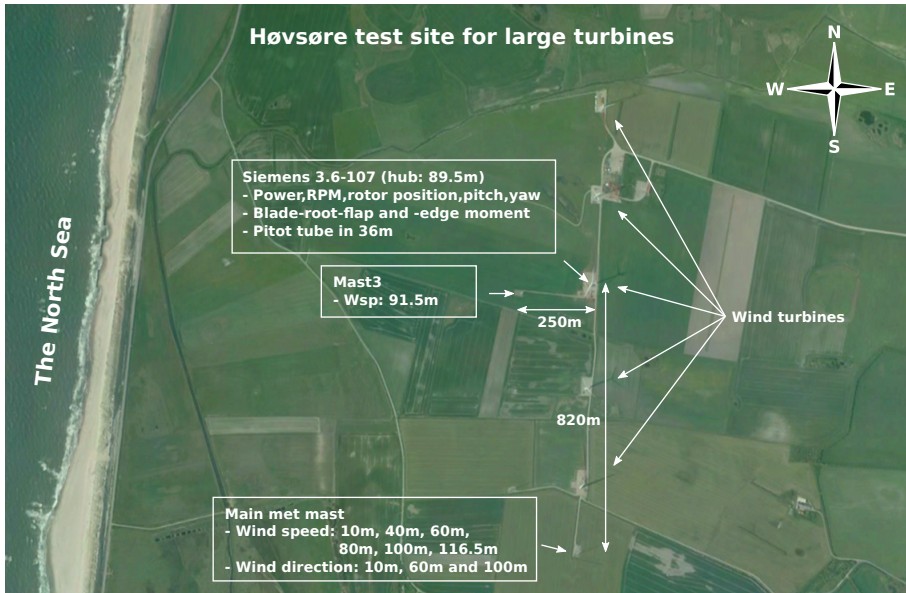

**Figure 3.** Overview of the Høvsøre Test Site for Large Wind Turbines in Denmark. The Siemens turbine is located in the middle of a row of five megawatt turbines.

based models for axial and tangential induction, a radial induction model and tip loss correction as well as models for skew and dynamic inflow.

Subtracting the estimated induction from the measured flow velocity results in an estimate of the free-stream inflow velocity that would have been observed at the same time and location without the presence of the turbine. In this step, uncertainty is also introduced due to the mismatch between the applied simple engineering models and the complex real world. The process and the introduced uncertainty are described in detail by Pedersen et al. (2018) that also, based on numerical simulations, conclude that the estimated free wind speed obtained from a BMFS is relatively accurate. Whether the introduced uncertainties outweigh the advantage of measuring at the blade will be investigated in this study.

### 2.3 Calibration of load sensors

The blade-root load sensors comprise flap- and edge-wise bending-moment sensors on all three blades. They are located 3.2 m from the hub centre.

A subset of the sensors is found to drift considerably with the temperature. A linear temperature correction is therefore applied before the calibration.

The edge-wise bending-moment sensors are calibrated using a set of time series measured at low wind speed and with pitch angles around 0°. In these cases, the edge-wise loads are dominated by the gravity loading, and the loads are therefore fitted to



a sinusoidal signal with magnitude equal to the own-weight moment of the blade:

$$\underset{a,b}{\text{Minimise}} \left( \sum_{\theta_{rotor}} |aMy(\theta_{rotor}) + b - M_{ow}\sin(\theta_{rotor})| \right) \tag{1}$$

where $a$ and $b$ are calibration factors, $My$ is the measured edge-wise bending moment, $\theta_{rotor}$ is the rotor position and $M_{ow}$ is the moment when the blade is in horizontal position due to the weight of the blade from the load sensor to the tip.

Similarly, the flap-wise bending-moment sensors can be calibrated using time series measured at a low wind and 90° pitch angle. The measurement database, however, contains no time series with 90° pitch and low wind, and it was therefore necessary to use time series with lower pitch angles for the calibration. Hence, the pitch angle must be included in the calibration formula:

$$\underset{a,b}{\text{Minimise}} \left( \sum_{\theta_{rotor}} |aMx(\theta_{rotor}) + b - M_{ow}\sin(\theta_{pitch})\sin(\theta_{rotor})| \right) \tag{2}$$

where $Mx$ is the measured flap-wise bending moment, and $\theta_{pitch}$ is the pitch angle.

The mean flap-wise bending moments of the three blades are not equal after this calibration. This is, however, justified as the measured pitch angles of blade 2 and 3 are offset by around -0.4 and +1° respectively, compared to blade 1. These pitch offsets are included in the simulations.

## 2.4    Derived tower load sensors

The current measurement database contains no tower-load sensors. The dynamic tower loads are, however, mainly induced by the aerodynamic blade loads, and it is therefore possible to derive tower-load estimations from the blade-root loads sensors.

The tower-bottom fore-aft bending moment is dominated by the constant weight of the rotor and the dynamic thrust on the rotor. The thrust is related to the rotor-plane projection of the blade-root bending moments (i.e. mainly the flap-wise bending moments) and using a linear calibration a good approximation can be achieved for a certain wind speed:

$$MTB_{foreaft,est} = a_{tb} \sum_{i=1..3} MBR_i + b_{tb} \tag{3}$$

where $MTB_{foreaft,est}$ is the estimated tower-bottom fore-aft bending moment, $MBR_i$ is the rotor-plane projection of the blade-root bending moment of blade $i$, and $a_{tb}$ and $b_{tb}$ are calibration constants.

Similarly, approximations of the tower-top tilt and yaw moments can be formulated:

$$MTT_{tilt,est} = a_{tilt} \sum_{i=1..3} MBR_i \cos\left(\theta_{rotor} - \frac{2i}{3}\pi\right) + b_{tilt} \tag{4}$$

$$MTT_{yaw,est} = a_{yaw} \sum_{i=1..3} MBR_i \cos\left(\theta_{rotor} - \frac{2i}{3}\pi + \frac{\pi}{2}\right) + b_{yaw} \tag{5}$$

The derived tower-load sensors have been calibrated based on HAWC2 simulations. Applied to other HAWC2 simulations with similar wind conditions, the tower loads derived from the blade-root sensors fit quite well with the actual simulated tower loads; see the example of 8 ms$^{-1}$ in Fig. 4.





The calibration constants are, however, dependent on the wind speed. Hence, the fine agreement seen in Fig. 4 is only obtainable when using the correct wind-speed-specific calibration constants.

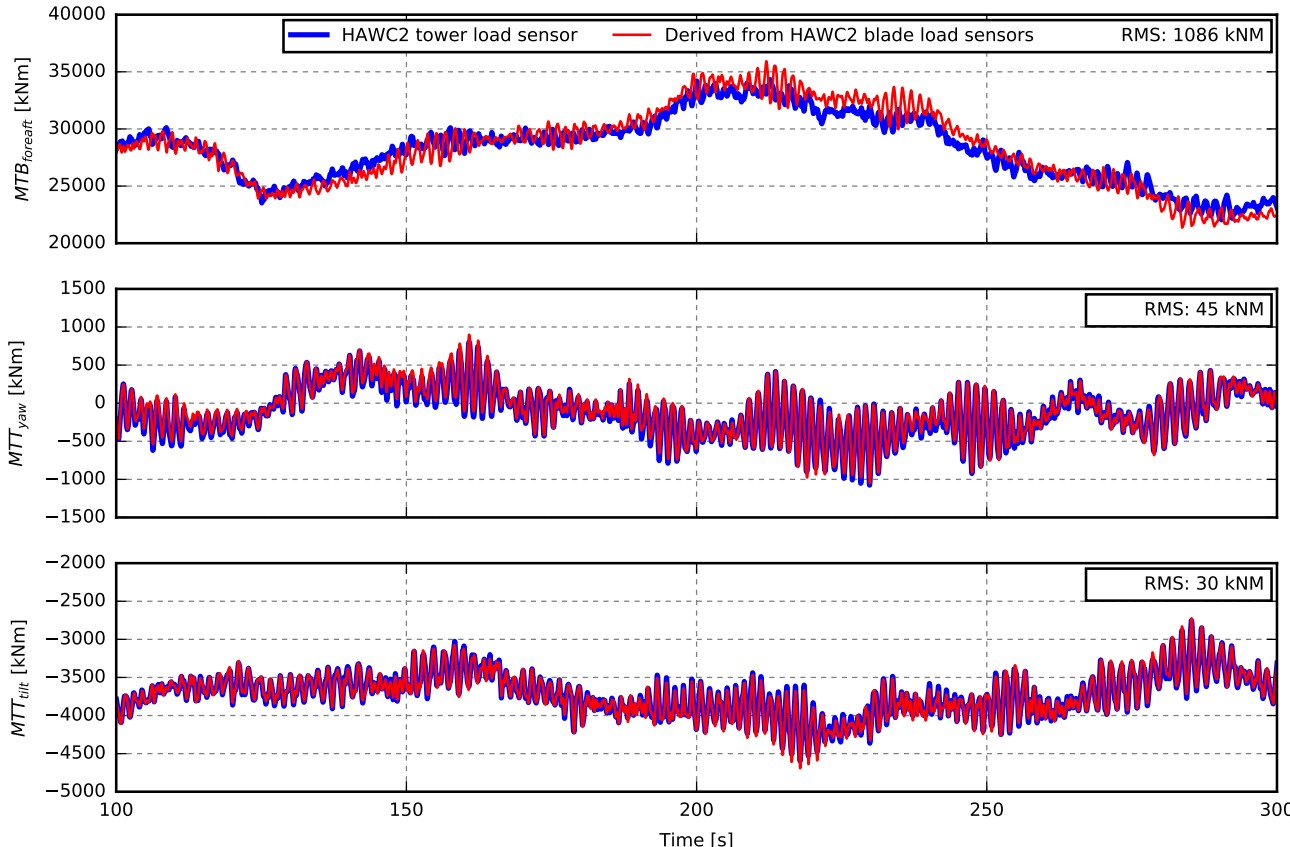

**Figure 4.** Comparison of the HAWC2 simulated tower loads and the derived tower loads, which is derived from the HAWC2 simulated blade-root load sensors and calibrated for 8 ms$^{-1}$.

The calibration constants are therefore determined for wind speeds ranging from 4 to 15 ms$^{-1}$ and interpolated based on the revolution-averaged pitot-tube mean wind speed. To test the calibration, the equivalent fatigue load of the derived tower-load sensors have been calculated for five independent simulation sets. The estimated loads are then compared to the HAWC2-simulated "real" tower loads. The relative error is shown in Fig. 5.

At low wind speeds, the tower-bottom bending moment is dominated by structural loads while the impact of the aerodynamic blade loads is limited. Hence, the derived tower-bottom sensor deviates considerably from the simulated tower-bottom signal, and the fatigue load error is relatively high; see Fig. 5. The derived tower-bottom fore-aft loads will therefore be discarded for wind speeds below 6 ms$^{-1}$. In all other cases, the mean error is less than 5 %. Note that this deviation will not affect the



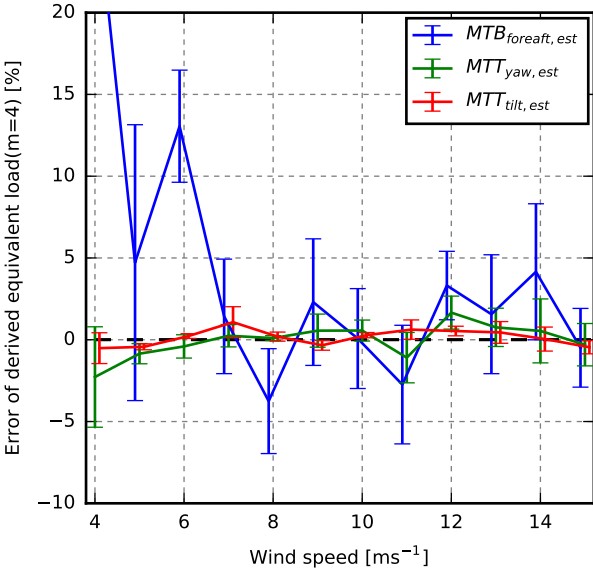

**Figure 5.** Relative fatigue load error of the derived tower load sensors compared to the HAWC2 simulated tower loads. The derived tower load sensors are obtained from the HAWC2 simulated blade-root load sensors and calibrated using wind-speed-dependent calibration constants.

discrepancies between the measurements and simulations in the results section directly, as the presented tower loads in both cases will be derived from the blade-root loads even though the "real" tower loads are also simulated directly by HAWC2.

## 2.5 Simulation model

The simulations used in this study are performed using HAWC2 - a non-linear aeroelastic code intended for computing wind turbine response in the time domain (Madsen et al., 2010a, 2012; Kim et al., 2013; Larsen et al., 2015).

The turbine model used for the simulations is based on the structural and aerodynamic data of the Siemens 3.6 MW turbine, which was tested at Høvsøre in 2009 during the DAN-AERO project; see Section 2.1. Within the HAWC2 framework, the turbine is controlled by the Basic DTU controller (Hansen and Henriksen, 2013), and the blades are modelled with slightly different pitch angles to match the offsets seen in the measurements; see Sect. 2.4.

## 2.6 Inflow characteristics

In this section, the inflow characteristics used for the different cases are described; see an overview of the 5 cases in Table 1. Cases 1 - 3 are based on met-mast sensors, while Cases 4 and 5 are based on the estimated free-stream pitot-tube wind speed; see Sect. 2.2. Table 2 gives an overview of the actual inflow parameters extracted from the 20 periods.





**Table 1.** Case overview.

| Case | Wsp | Wsp trend | Tint | Shear | $L, \Gamma$ | $\alpha\epsilon$ fitted to | Constrained to |
|------|-----|-----------|------|-------|-------------|----------------------------|----------------|
| Case 1 | Mast3 | - | Site avg. | Site avg. | Standard | - | - |
| Case 2 | Mast3 | Mast3 | - | Main met mast | Stability dependent | Mast3 variance | - |
| Case 3 | Mast3 | Mast3 | - | Main met mast | Stability dependent | Mast3 variance | Mast3 wsp |
| Case 4 | Pitot | Pitot | - | Pitot (power-law) | Standard | Pitot variance | - |
| Case 5 | Pitot | Pitot | - | Pitot (grid) | Standard | Pitot variance | Pitot wsp |

**Table 2.** Inflow characteristics of P1 - P20.

| | | Wsp $[\mathrm{ms}^{-1}]$ | | Trend $\left[\frac{\mathrm{ms}^{-1}}{10\mathrm{min}}\right]$ | | Turb. int. [%] | | | Power shear exp. [-] | | | Stability |
|---|---|---|---|---|---|---|---|---|---|---|---|---|
| Obtained from | | M[1] | P[2] | M[1] | P[2] | S[3] | M[1] | P[2] | S[3] | M[1] | P[2] | M[1] |
| P1 | 2009-07-02 05:30 | 8.1 | 7.9 | -0.2 | -0.2 | 7.8 | 3.5 | 2.8 | 0.09 | 0.21 | 0.15 | Stable |
| P2 | 2009-07-05 17:10 | 8.0 | 7.5 | -0.2 | 0.2 | 7.8 | 3.1 | 3.4 | 0.09 | 0.08 | 0.03 | Very unstable |
| P3 | 2009-05-10 19:00 | 8.0 | 8.2 | -0.2 | 0.1 | 7.8 | 5.3 | 3.0 | 0.09 | 0.09 | -0.01 | Unstable |
| P4 | 2009-07-05 08:50 | 8.1 | 7.5 | -0.3 | -0.4 | 7.8 | 7.1 | 5.9 | 0.09 | 0.09 | 0.00 | Very unstable |
| P5 | 2009-07-05 14:30 | 7.9 | 7.5 | 0.1 | 0.5 | 7.8 | 9.2 | 7.0 | 0.09 | 0.06 | 0.01 | Very unstable |
| P6 | 2009-07-05 09:10 | 8.0 | 7.6 | -0.6 | -0.7 | 7.8 | 5.2 | 6.1 | 0.09 | 0.06 | -0.00 | Very unstable |
| P7 | 2009-07-05 02:10 | 8.1 | 7.8 | -2.9 | -1.6 | 7.8 | 7.0 | 6.1 | 0.09 | 0.13 | 0.02 | Neutral |
| P8 | 2009-05-10 10:30 | 7.9 | 8.1 | 0.9 | 1.3 | 7.8 | 8.1 | 6.0 | 0.09 | 0.07 | -0.01 | Very unstable |
| P9 | 2009-07-05 00:10 | 8.1 | 7.5 | 1.3 | 2.0 | 7.8 | 7.0 | 7.3 | 0.09 | 0.12 | 0.01 | Unstable |
| P10 | 2009-05-24 20:50 | 8.1 | 7.9 | -1.7 | -1.9 | 7.8 | 6.9 | 7.7 | 0.09 | 0.10 | -0.00 | Very unstable |
| P11 | 2009-07-09 04:50 | 13.8 | 13.7 | 0.6 | 0.4 | 7.2 | 5.6 | 4.6 | 0.13 | 0.13 | 0.03 | Very unstable |
| P12 | 2009-07-09 02:20 | 13.9 | 13.6 | -0.6 | -0.5 | 7.2 | 5.8 | 4.8 | 0.13 | 0.12 | 0.04 | Near unstable |
| P13 | 2009-05-23 03:30 | 14.3 | 14.4 | 1.6 | 2.2 | 7.2 | 8.0 | 6.5 | 0.13 | 0.13 | 0.09 | Unstable |
| P14 | 2009-05-23 00:40 | 13.8 | 13.5 | -1.0 | 0.2 | 7.2 | 7.2 | 6.9 | 0.13 | 0.13 | 0.07 | Near unstable |
| P15 | 2009-05-09 01:30 | 13.8 | 13.6 | 0.6 | 0.5 | 7.2 | 6.9 | 6.3 | 0.13 | 0.15 | 0.07 | Neutral |
| P16 | 2009-05-23 02:00 | 14.1 | 13.4 | -1.2 | -0.7 | 7.2 | 5.7 | 6.6 | 0.13 | 0.13 | 0.11 | Near unstable |
| P17 | 2009-05-09 01:10 | 14.0 | 13.4 | 0.0 | 0.1 | 7.2 | 6.9 | 6.6 | 0.13 | 0.15 | 0.06 | Neutral |
| P18 | 2009-05-09 01:40 | 14.0 | 13.6 | -0.4 | 1.0 | 7.2 | 6.3 | 5.2 | 0.13 | 0.15 | 0.07 | Neutral |
| P19 | 2009-07-09 03:50 | 14.2 | 13.8 | 1.2 | 1.5 | 7.2 | 5.8 | 7.0 | 0.13 | 0.12 | 0.03 | Unstable |
| P20 | 2009-05-23 02:40 | 14.0 | 13.8 | 1.4 | 0.2 | 7.2 | 8.3 | 7.6 | 0.13 | 0.13 | 0.12 | Near unstable |

[1] Mast 3 / main met mast

[2] Pitot tube

[3] Mast 3 / main met mast (Site average)



### 2.6.1   Wind speed

In Cases 1 - 3, the 10-min-mean wind speed measured at Mast3 is used. Mast3 is located around 2.5 diameters to the west of the turbine; see Fig. 3. Its 10-min-mean wind speed is therefore expected to match the mean wind speed at the rotor quite well in the selected periods.

In Cases 4 and 5, the mean wind speed is extracted from the estimated free-stream pitot-tube wind speed. To avoid the problem that the mean wind speed is influenced by non-linear shear, only observations recorded in 85 - 95 m are included (i.e. the hub height $\pm 5$ m at both sides of the rotor).

### 2.6.2   Wind speed trend

In some of the selected periods, the mean wind speed changes considerably during the period. A linear wind-speed trend is
therefore calculated for all periods and included in the simulations in all cases except Case 1.

Wind speed trends may result in increased loads, e.g. tower-bottom fatigue loads, as the trend will contribute with one (large) cycle. Furthermore, the target turbulence intensity will be too high if calculated from the standard deviation of the raw wind speed signal. Note, however, that periods with wind speed trends may be problematic as it means that the turbulence conditions are not stationary, and the theory behind the applied turbulence model assumes stationary conditions.

### 2.6.3   Shear and mean wind speed variation

The wind shear profile has a high impact on the flap loads as well as on the tower-top tilt and yaw loads. The 10-min-mean wind speed is not known in all parts of the rotor, and therefore a shear model is necessary. In this study, the power-law shear profile is used, and it is fitted to one hour of measurements. As the wind may change during one hour, we would like to base the shear profile on the selected 10-min observations. The 10-min-mean vertical profile, however, can have almost any shape,
and therefore a longer time period is usually required to make a proper power-profile fit.

In Case 1, the site-average wind-speed-dependent shear profile is used while the mean wind speeds at different heights, measured at the main met mast 850 m away, are used to estimate the vertical shear profile for Cases 2 and 3. Note that the main met mast has sensors up to 116.5 m, and the upper part of the rotor is therefore not represented.

It is possible to use the measured 10-min pitot-tube shear profile directly inside the pitot tube altitude range, but another
approach is required outside this range. A power-law shear profile is therefore fitted to one hour of the estimated free-stream pitot-tube wind speed and used for Cases 4 and 5.

Ideally the 10-min-mean wind speed is known everywhere at the rotor. This is obviously not the case, but from the pitot tube measurements, the 10-min-mean wind speed at the path of the pitot tube can be extracted and used to specify the mean wind speed in a grid covering the rotor; see Fig. 6 (left). This information is used in combination with the one-hour power-shear
profile (Fig. 6 (middle)) to specify a grid-based mean wind speed for Case 5; see Fig. 6 (right).





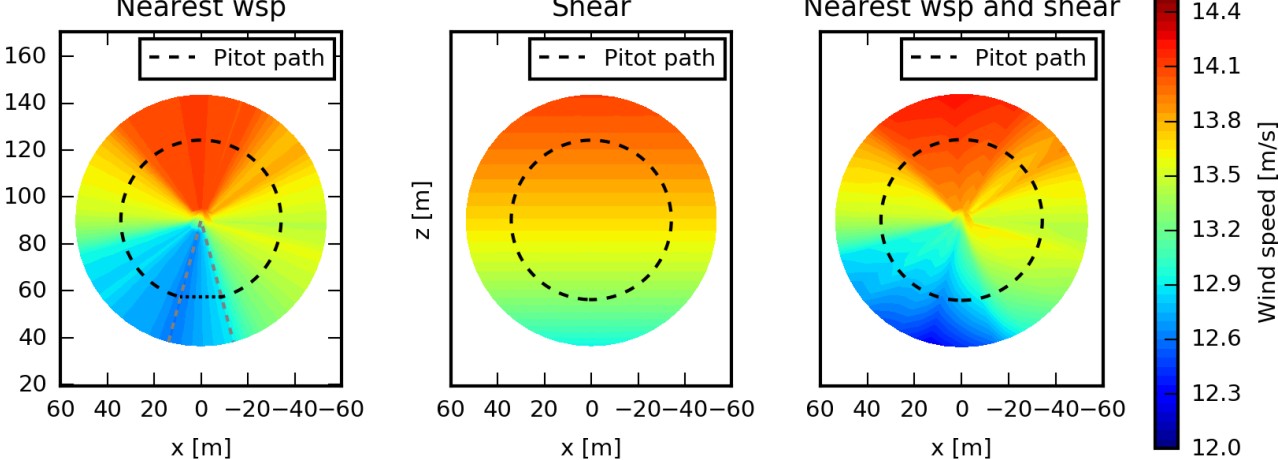

**Figure 6.** Left: wind speed based on nearest pitot-tube wind speed (interpolated values are used in a 30° sector around the tower to the exclude effects of tower shadow). Middle: wind speed based on the one-hour power-shear profile. Right: wind speed based on the nearest pitot-tube wind speed and power shear profile.

The aerodynamic models that are used to estimate the free-stream pitot-tube wind speed do not include a model of the tower shadow. The wind speed drop due to tower shadow should not, however, be included in the inflow input to the simulations. The mean wind speed is therefore linearly interpolated in a 30° sector around the tower as indicated in Fig. 6 (left).

### 2.6.4 Turbulence

5 The turbulence used in the simulations is generated using the Mann turbulence model (Mann, 1994, 1998). This model requires three parameters as input: a length scale of the spectral velocity tensor, $L$, an energy dissipation factor, $\alpha\epsilon$, and a shear distortion parameter, $\Gamma$. Standard parameters can be used, or they can be fitted to the turbulence spectra calculated from a long period of e.g. 3D sonic measurements.

For Cases 1, 4 and 5, standard values are used for $L$ and $\Gamma$ as specified in IEC 61400-1 (2005) while fitted values are used 10 for Cases 2 and 3.

The Mann turbulence model assumes neutral atmospheric stability conditions. The parameters can, however, be fitted to spectra non-neutral stability classes where slightly different parameters are obtained. The stability-dependent parameters used for Cases 2 and 3 (see Table 3) are extracted from Peña et al. (2010) who investigated the turbulence at the current site.

Standard- or long-term-average values may be appropriate for $L$ and $\Gamma$, but as we want to simulate the current situation and 15 not a monthly or yearly average, another approach is required for the $\alpha\epsilon$-parameter that for fixed $L$ and $\Gamma$ is closely related to the turbulence intensity and, thus, the fatigue loads.





**Table 3.** Standard and stability dependent turbulence parameters.

|  | Length scale, $L$ | Shear distortion, $\Gamma$ |
|---|---|---|
| Standard (IEC) | 33.6 | 3.9 |
| Very stable | 7.7 | 2.88 |
| Stable | 11.6 | 2.79 |
| Near stable | 24.6 | 2.68 |
| Neutral | 33.1 | 2.57 |
| Near unstable | 50.8 | 3.32 |
| Unstable | 69.2 | 2.09 |
| Very unstable | 79.1 | 1.54 |

In Case 1, the turbulence is scaled after generation, such that the turbulence intensity in the centre of the turbulence field matches the turbulence intensity measured by Mast3 in the selected period. This approach is convenient as it ensures agreement between the measured and simulated hub-height turbulence intensity. It may, however, result in energy from scales that are not represented in the turbulence model being distributed on other frequencies. Furthermore, the approach is inappropriate if the

centre of the turbulence field is not representative for the whole field.

In Cases 2 - 5, the $\alpha\epsilon$ parameter is defined, such that the integral of the $uu$-Mann-model spectrum equals the integral of the measured $uu$ spectrum. For Cases 2 and 3, the measured $uu$ spectrum is obtained from the detrended wind speed measured by Mast3, while the pitot-tube-based wind speed is used for Cases 4 and 5.

Due to the low fixed-position resolution of the pitot-tube wind speed, only the low frequency part of the $uu$ spectrum can

be obtained from the pitot tube, and this part is not suitable for fitting. Assuming that the turbulence field is homogeneous, the $uu$ spectra are therefore calculated from all of the pitot tube observations after subtracting the position-dependent mean wind speed and trend. The resulting spectra are very different from the normal fixed-position spectra because the pitot tube moves in and out of turbulence structures, as also reported by Hardesty et al. (1981) and Verholek (1978), and theoretically described by Kristensen and Frandsen (1982). The variance of the turbulence, i.e. the integral of the spectrum, is, however, independent

of the frame of reference.

In Cases 3 and 5, the measured wind speeds are used as the input to a constraint turbulence simulator that modifies existing turbulence fields, e.g. stochastic realizations of the Mann turbulence model, to reproduce the specified wind speeds at the corresponding positions while preserving the statistics. The applied constraint turbulence simulation approach is described by Nielsen et al. (2003). In Case 3, the wind speed measured by Mast3 is used to constrain the turbulence at the position of Mast3,

while the pitot-tube wind speed is used to constrain the turbulence in Case 5 at the instantaneous position of the rotating pitot tube.



# 3 Results

Figures 7, 8 and 9 show the equivalent loads coloured by turbulence intensity, shear and atmospheric stability, respectively. The strongest dependence on these three single parameters is seen by the tendency to stratification in the flap and tower-bottom loads for low wind speeds, where the lowest loads are seen to occur in stable conditions with low turbulence intensity and high

5    shear. The colours are, however, rather mixed, and wide areas have similar colours. It is therefore concluded that the scatter is somewhat independent of these three single parameters, and a more sophisticated approach, which considers the actual combination of inflow parameters, is required to predict the loads of specific periods.

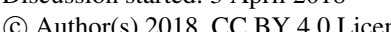

**Figure 7.** 1 Hz equivalent loads coloured by the turbulence intensity measured at Mast3.



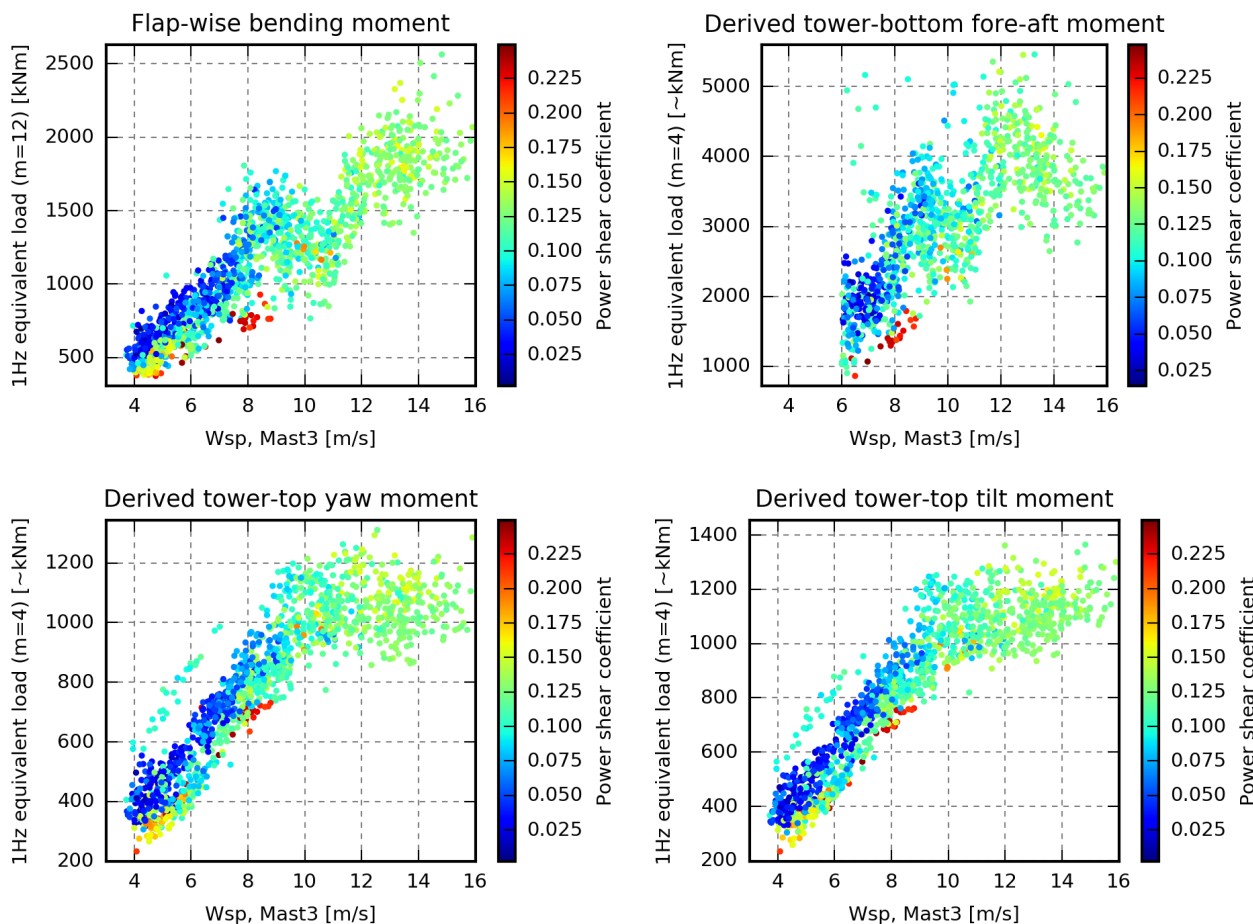

**Figure 8.** 1 Hz equivalent loads coloured by the power shear coefficent extracted from the main met mast.




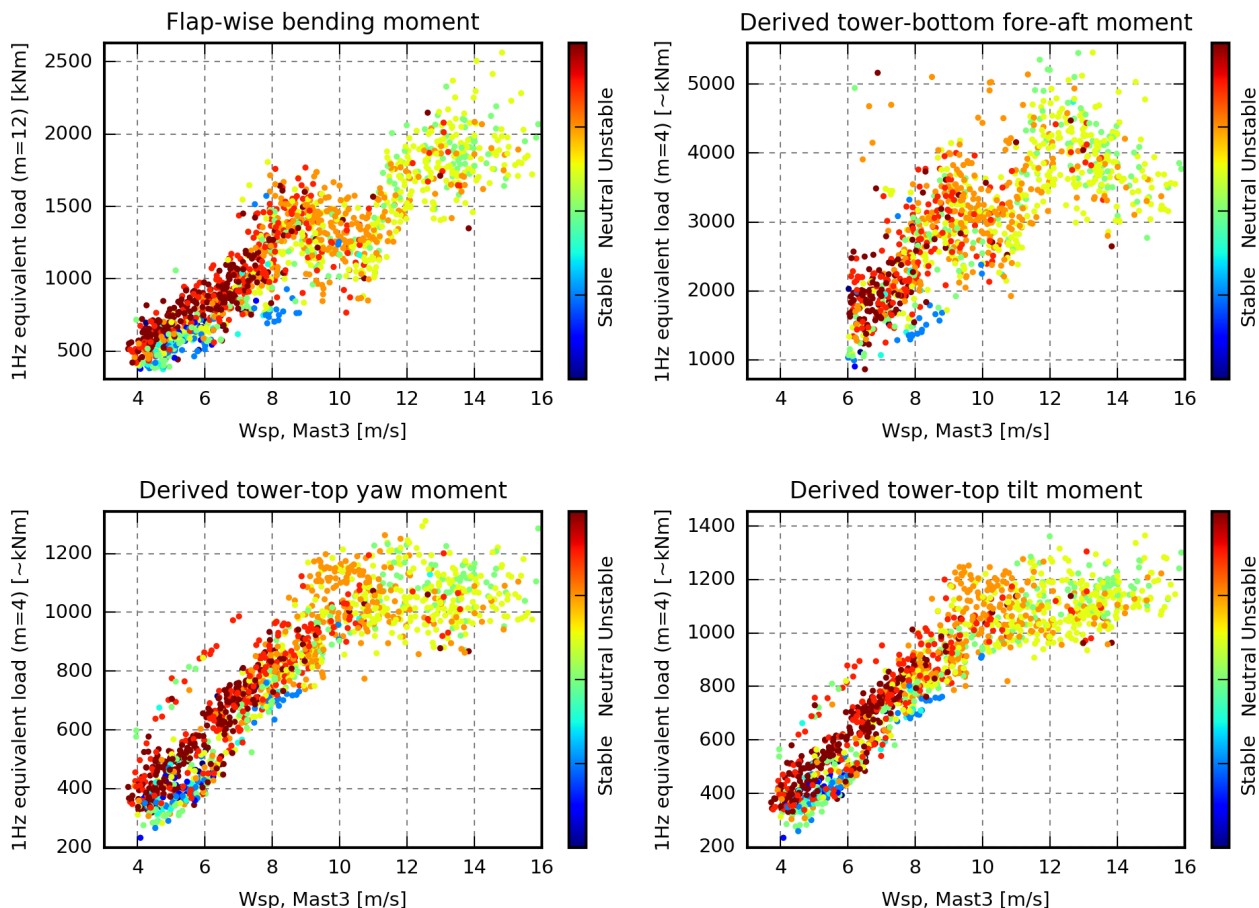

**Figure 9.** 1 Hz equivalent loads coloured by atmospheric stability extracted from the main met mast.




An overview of the mean absolute relative error of the different cases can be found in Fig. 10, while Fig. 11 shows the distribution of the relative simulation errors. Figure 12 shows how to interpret Fig. 13 - 16, which offer more details of the cases by showing the measured and simulated loads of P1 - P20.

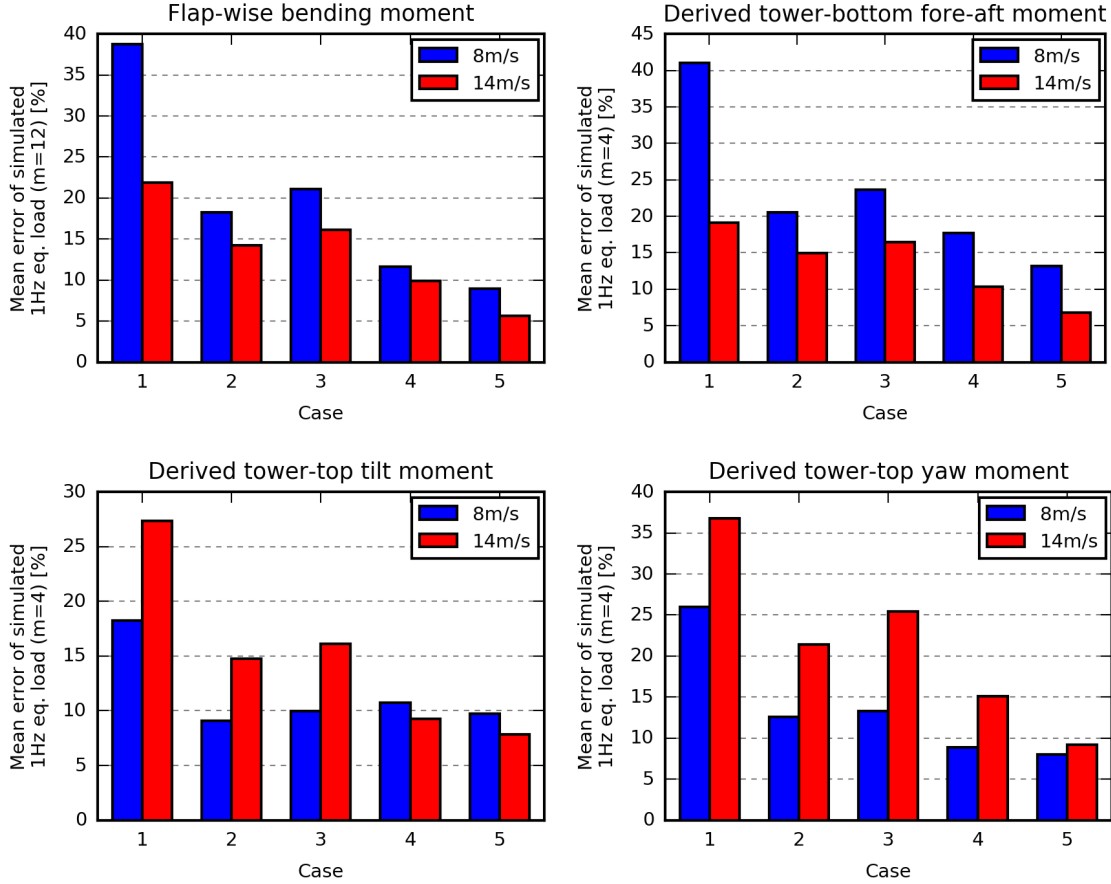

**Figure 10.** Mean absolute error of the simulated equivalent loads.



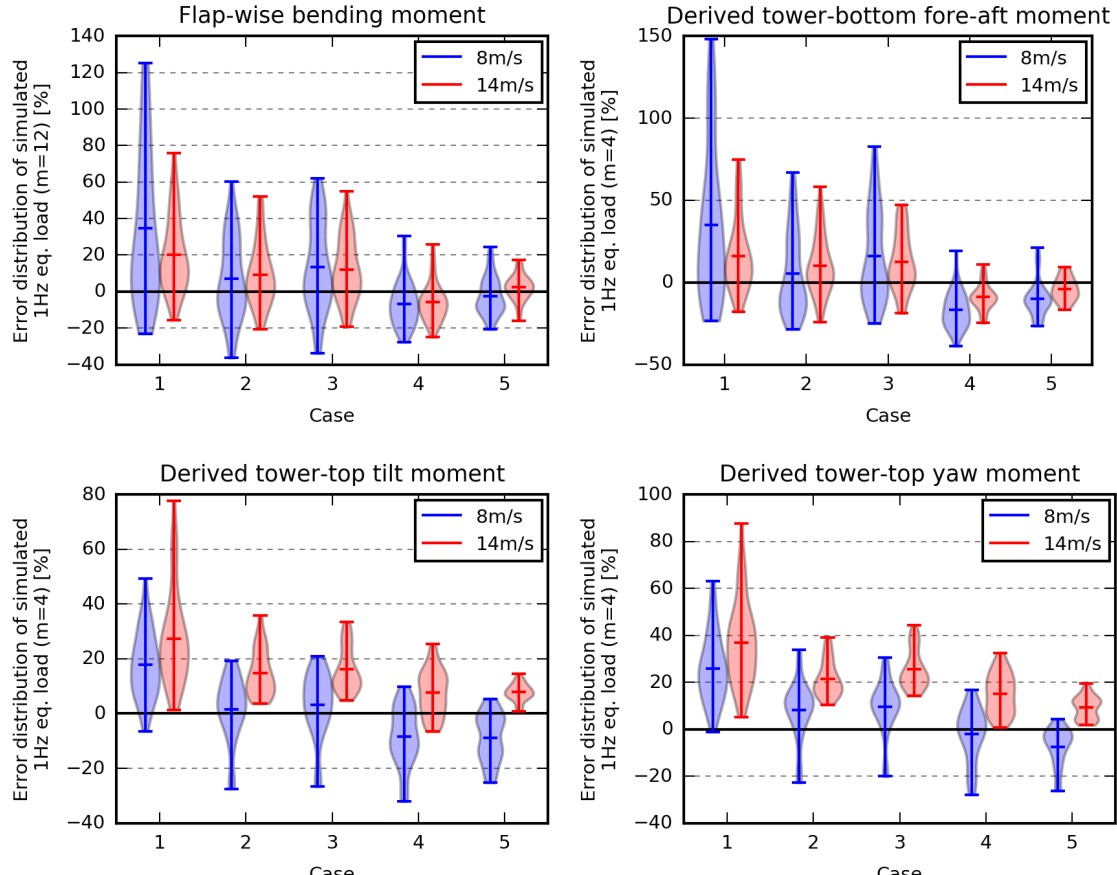

**Figure 11.** Error distribution of the simulated equivalent loads.



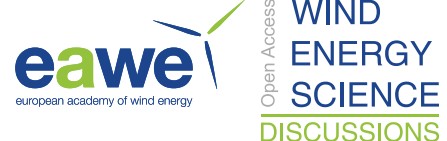

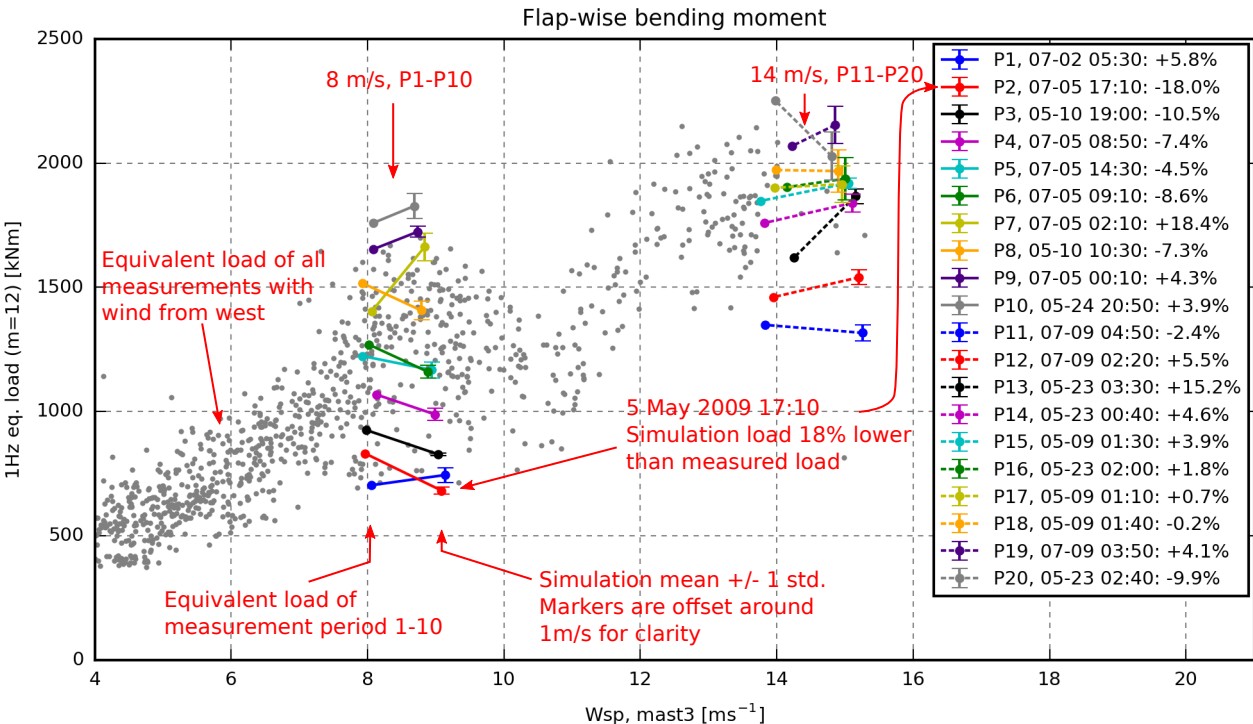

**Figure 12.** Example showing how to interpret Fig. 13 - 16. The title states that the figure shows the equivalent flap-wise bending moment of blade A. The grey background dots represent the equivalent loads of all measurements with wind from the west; i.e. no-wake situations. The 20 selected periods, P1 - P10 at 8 $\mathrm{ms}^{-1}$ and P11-P20 at 14 m/s, are illustrated by dots connected to error bars. The dots show the measured equivalent load and wind speed while the error bars illustrate the simulated mean loads $\pm 1\sigma$. Note that the error bars are offset around 1 $\mathrm{ms}^{-1}$ to the right for clarity. The red dot and error bar, for instance, represent P2; i.e. 7th May, 17:10 - 17:20. The equivalent load measured in this period was around 830 kNm, while the six corresponding simulations have a mean load level around 682 kNm and a standard deviation of 16 kNm.





In **Case 1**, only the wind speeds are different between the periods. The load levels within the two wind-speed groups are therefore very similar as seen in Fig. 13. In this case, the simulated loads do not reflect the measured load variation, and the mean absolute relative error seen in Fig. 10 is therefore high, especially for the flap and tower-bottom loads at $8\,\mathrm{ms}^{-1}$ where the relative variation is huge, but also in the tilt and yaw moments at $14\,\mathrm{ms}^{-1}$ where the simulated loads are too high. It should

5   also be noted that the variation of the simulations due to different turbulence realisations (seeds) does not reflect the measured variation, except for the yaw and tilt moment in the high-wind situations.




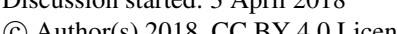

**Figure 13.** Case 1. Site average turbulence intensity and shear (wind speed trend neglected). For interpretation see Fig. 12.





In **Case 2**, information about the wind speed trend, the measured turbulence level and shear profile is included in the simulations.

Including the wind speed trend increases the loads considerably in some periods. In P7, for example, the mean wind speed decreases 2.9 $\mathrm{ms}^{-1}$ (linear fit) during the 10 minutes; see Table 2. Including this trend increases the flap and tower-bottom fatigue loads by around 30 %. It indicates that wind-speed trends are important to include in simulations for load validations.

In the selected periods, the turbulence intensity varies from 3.1 to 9.2 %. Including this information makes the range of the simulated loads reflects the range of the measured loads. The turbulence scaling approach, which is used for Case 1, is found to introduce substantial variation due to different turbulence realisations (seeds). This variation is considerably reduced in this and the succeeding cases by fitting the $\alpha\epsilon$ turbulence parameter. Seen in isolation, the $\alpha\epsilon$-fitting method reduces the average seed-induced variation of yaw loads at 14 $\mathrm{ms}^{-1}$ from 450 kNm to 90 kNm, while the maximum error of the tilt and yaw moments at 14 $\mathrm{ms}^{-1}$ is approximately reduced from 80 % to 40 %.

The terrain is rather flat towards the west, and the power-shear exponents are therefore modest (0.06 to 0.21), and in general similar to the site-average values (0.09 for 8 $\mathrm{ms}^{-1}$ and 0.13 for 14 $\mathrm{ms}^{-1}$). The largest difference is found in P1, where the shear coefficient is increased from 0.09 to 0.21, which seen in isolation increases the simulated flap loads of this period by 9-18 %. In general, however, the effect of including the measured shear profile is limited, but the situation may be different if periods with wind from other directions were also considered.

Furthermore, the stability dependent $L$ and $\Gamma$ parameters are used for the turbulence generation. Using these non-standard parameters, affects the flap and tower-bottom loads significantly in some periods. In P1 (stable conditions), the tower-bottom load decreases by 22 %, while it increases by 20 % in P11 (very unstable conditions). In these periods, however, the error of the simulated loads is not reduced.

Figure 10 reveals that the mean error of all loads is significantly reduced by utilising these inflow characteristics. The correlation between the measured and simulated load levels is, however, still poor. The simulated tower-bottom load of P5, for instance, is up to 67 % too high, and the measured tilt-moment fatigue loads of P2 and P5 are almost equal, but they account for the minimum and maximum simulated loads, respectively; see Fig. 14





**Figure 14.** Case 2. Best case based on met mast inflow information. For interpretation see Fig. 12.





In **Case 3**, constraint turbulence simulation has been applied to constrain the turbulence to match the Mast3 wind speed at the position of Mast3, i.e. 250 m upstream. It has an effect on most of the simulated loads, but it slightly increases the mean error of all load sensors; see Fig. 10.

The biggest error increase is seen for P5, which has a distinct drop in the wind speed measured by Mast3 in the middle of the period. In the simulations, a similar drop, introduced by the constraint turbulence simulator, is transported unaffected with the steady mean wind to the turbine in agreement with Taylor's frozen turbulence hypothesis (Taylor, 1938). Around 30 s later, the same wind speed drop therefore hits the turbine and induces significant fatigue loads. In the real world, however, the turbulence structures changes, the mean wind is not always steady, and the wind-speed drop may even pass beside the turbine. In P5, a small drop is measured in the flap-wise bending moment, but it is only half the size of the simulated drop.

**Case 4** uses inflow characteristics extracted from the estimated free-stream pitot tube wind speed. As seen in Table 2, these characteristics are different from the met mast characteristics; the mean wind speed deviates up to 0.77 ms$^{-1}$, the wind speed trend up to 1.45 ms$^{-1}$, the turbulence intensity up to 2.3 % and the power shear coefficient up to 0.11.

The mismatches are caused by the spatial distance between the locations of measurements, fundamental differences in the sensor technology and measurement method, and the uncertainties introduced in the conversion from pitot-tube measurement to free-stream wind speed in the fixed global coordinates; see section 2.2.

Compared to Case 2 (the most equivalent met-mast case), all mean errors decrease by 5 % or more except the mean error of the tilt moment at 8 ms$^{-1}$; see Fig. 10. The error ranges also decrease considerably for the flap and tower-bottom loads (see Fig. 11), while they are similar for the tilt- and yaw-moment error.



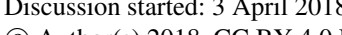



**Figure 15.** Case 4. Best case based on pitot tube inflow information. For interpretation see Fig. 12.



In **Case 5**, the measured mean-wind-speed variations within the rotor is modelled, and furthermore, the instant measured pitot tube wind speed is used to constrain the turbulence model.

Modelling the measured mean-wind-speed variations via the grid-based approach (exemplified in Fig. 6) increases all loads, except the yaw moments. In some periods, the flap load increases up to 15 %, and seen in isolation, the use of this approach slightly decreases the error of most of the simulated loads. It may be that the mismatch introduced by extrapolating the wind speed measured on the pitot tube path to the whole rotor area almost neutralises the positive effects, in which case more pitot tubes would be beneficial.

In this case, the turbulence field is generated using standard $L$ and $\Gamma$ parameters and constraint turbulence simulation. In theory, this approach is problematic as the statistics of the applied constraints may be different from the standard parameters, such that the constraint turbulence simulator needs to compensate in other parts of the turbulence field to obtain the requested statistics. Using the stability-dependent $L$ and $\Gamma$ parameters instead has been tried. It was found to have a small positive effect on the errors at $8\,\mathrm{ms}^{-1}$ and a similar small, but negative, effect on the errors at $14\,\mathrm{ms}^{-1}$. We have therefore chosen to use the standard parameters in this case, to avoid the need for met-mast measurements to determine the stability conditions.

In the selected periods, the use of constraint turbulence simulation reduces the mean error for all load sensors. Furthermore, the range of the simulated loads due to different turbulence realisations decreases considerably, such that the need for multiple simulations with different seeds is reduced; see Fig. 16.

In Cases 5, the range of the simulated loads reflects the range of the measured loads. They are therefore assumed to be much more suitable for load extrapolation than the loads of Case 1.

The derived tower loads are slightly underestimated at $8\,\mathrm{ms}^{-1}$ and overestimated at $14\,\mathrm{ms}^{-1}$. These deviations may be introduced by the tower-load derivation and calibration procedure, uncertainties in the measured pitch angle offsets, and by different control behaviour due to differences between the Siemens controller and the Basic DTU controller.

Only a few of the lines that connect the measured and simulated flap and tower-bottom observations intersect, meaning that the inflow conditions that result in high load levels in the measurements also result in high load levels in the simulations and vice versa. The same tendency is seen for the tilt and yaw moment at $14\,\mathrm{ms}^{-1}$.







**Figure 16.** Case 5. Best case based on pitot tube inflow information. For interpretation see Fig. 12.





At the beginning of this section, it was concluded that an advanced approach that considers combinations of inflow parameters would be required to predict the loads of specific periods. Aeroelastic simulations can be considered to be such an approach, and to compare to the single parameter approach in Fig. 7, 8 and 9, two additional simulation sets were performed. Both sets comprise 970 simulations representing all suitable periods in the measurement database (one seed per period). In the first set, inflow information is extracted from the met masts (similar to Case 2) while the second set is based on information from the pitot tube (similar to Case 5). Figure 17 and 18 show the equivalent loads, coloured by the HAWC2-simulated load relative to the wind-speed-dependent measured load range. This means that the red dots represent periods where the simulated load equals the maximum measured load at that wind speed, while the blue dots represent periods where the simulated load equals the minimum measured load. In other words, unmixed rainbow-coloured scatter means that the measured and simulated loads are similar and that the measured scatter can be predicted.

The most promising result is seen in the flap and tower-bottom loads coloured by the pitot-tube-based simulations (top row of Fig. 18) where the scatter is almost rainbow-coloured. This means that HAWC2 simulations with inflow characteristics extracted from the pitot tube are able to explain most of the measured flap and tower-bottom load scatter. The met-mast-based counterparts (top row of Fig. 17) are more mixed, even though most of the red observations are in the upper part of the scatter and most of the blue observations are in the lower part.

The tilt and yaw moment scatter, on the other hand, cannot be explained using these approaches. In both cases, most high-load observations are underestimated from 4 to 8 $\mathrm{ms}^{-1}$ and from 10 to 12 $\mathrm{ms}^{-1}$, while low-load observations are overestimated from 8 to 10 $\mathrm{ms}^{-1}$ and above 12 $\mathrm{ms}^{-1}$.



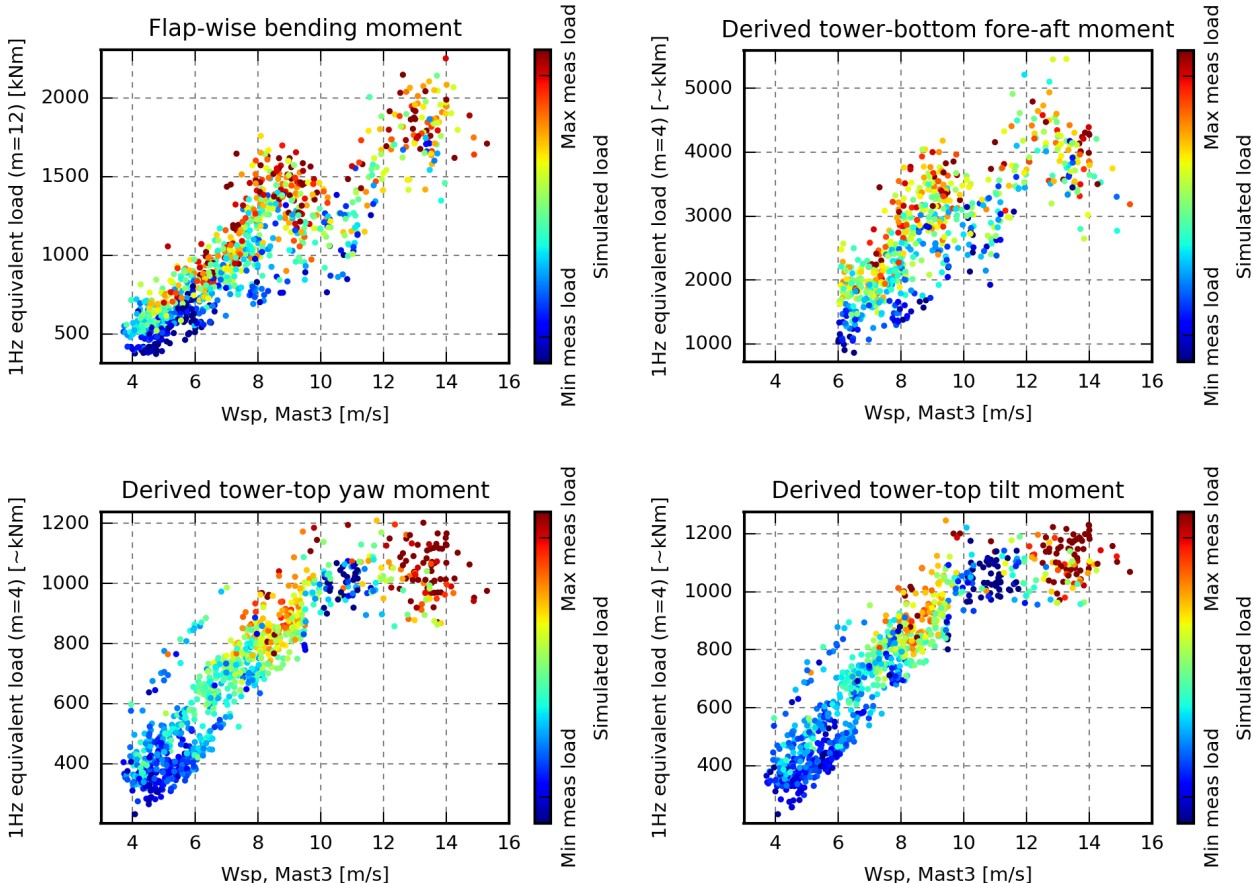

**Figure 17.** Equivalent measured loads, coloured by the corresponding simulation result. The simulations are performed using inflow information from the met masts, similar to Case 6 (but only one seed per period). If the simulated load equals the maximum measured load at the current wind speed, then the observation is red, while observations where the simulated load equals the minimum load measured at the current wind speed are blue.



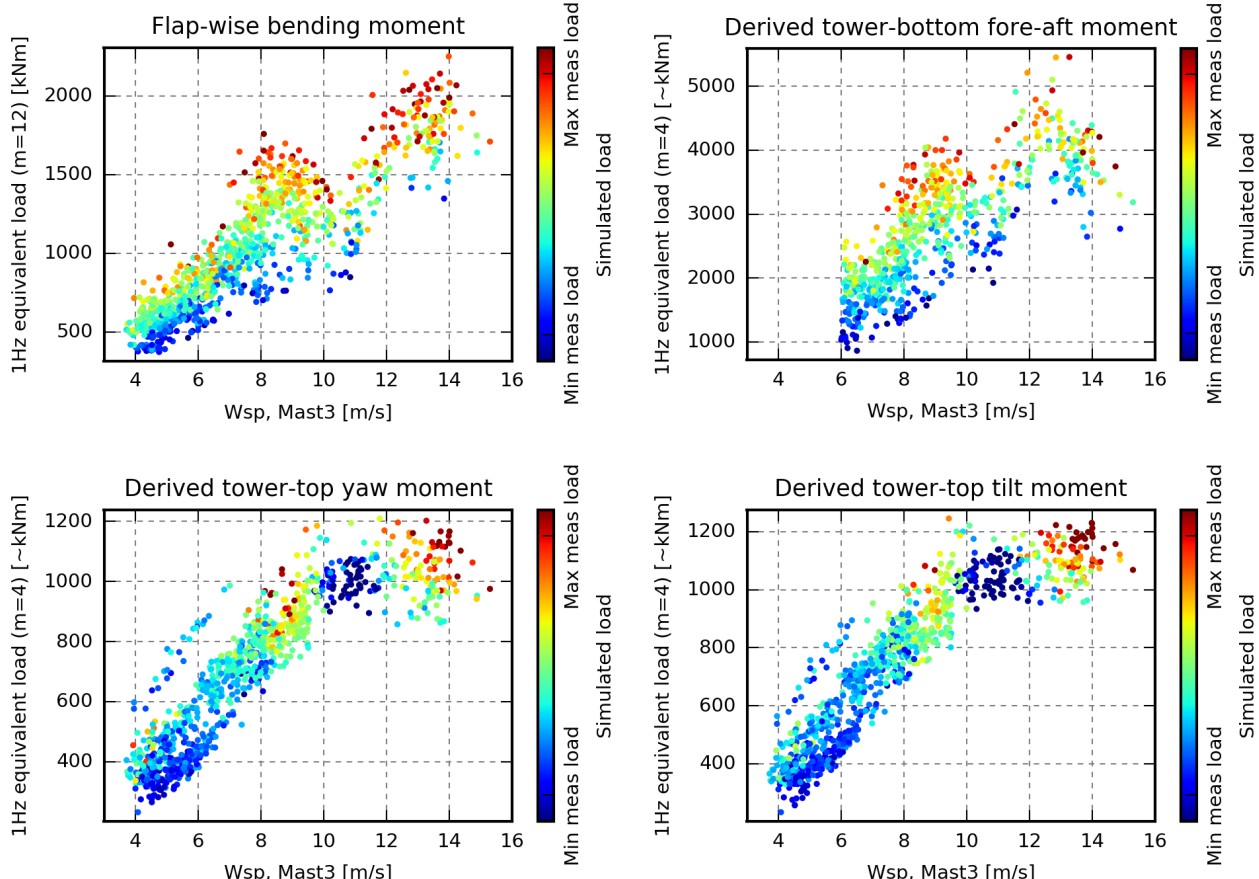

**Figure 18.** Equivalent measured loads, coloured by the corresponding simulation result. The simulations are performed using inflow information from the pitot tube, similar to Case 11 (but only one seed per period). If the simulated load equals the maximum measured load at the current wind speed, then the observation is red, while observations where the simulated load equals the minimum load measured at the current wind speed are blue.





## 4   Conclusions

In this paper, different inflow information is extracted from a measurement database and used for aeroelastic simulations to investigate if using more detailed inflow descriptions improves the accuracy of the simulated loads.

The inflow information is extracted from nearby met masts and a blade-mounted five-hole pitot tube. The pitot tube is located
inside the induction zone, i.e. the measured flow velocity is influenced by the presence of the turbine. An aerodynamic model is therefore used to estimate the free-stream inflow velocity that would have been observed at the position of the pitot tube without the presence of the turbine.

In the case study, 20 periods, which represent a wide range of loads at 8 and 14 $\mathrm{ms}^{-1}$, were selected. From these periods, inflow information was extracted for simulations.
The case study revealed that the loads in simulations based on site-average turbulence intensity and shear profile (the typical load validation approach) did not reflect the measured loads, and most of the simulated load ranges were considerably smaller than the ranges of measured loads. Load extrapolation based on this approach may therefore be misleading.

Including the met-mast measured turbulence intensity increases the variation of the simulated loads, and makes the simulated load range reflect the measured range. The one-to-one correspondences were, however, poor, with deviations up to 67 %.
The turbulence scaling approach, where the turbulence is scaled such that the turbulence intensity in the centre of the field matches the target intensity, was found to introduce a considerable variation in the simulated loads. Therefore, the scaling the turbulence such that the integral of the target $uu$-spectrum matches the target variance is highly recommended.

In most periods, the inflow characteristics extracted from the pitot tube deviate from the inflow characteristics extracted from the met masts. The mismatches are caused by the spatial distance between the locations of met masts and the pitot tube,
fundamental differences in the sensor technology and measurement method, and uncertainties introduced in the conversion from pitot tube measurement to estimated free-stream inflow wind speed in fixed global coordinates.

Using the wind speed, turbulence intensity and shear measured by the blade-mounted pitot tube reduces the errors of the flap and tower-bottom loads in this study, while the errors of the tilt and yaw moments are similar. This indicates that it is beneficial to measure the inflow with a BMFS even though errors are introduced due to the dynamic and static deflection and torsion of
the blade, as well as in the aerodynamic model that corrects for the turbine induction.

Including the measured wind speed trend, shear profile, rotor-position-dependent variations in the mean wind, and stability-dependent turbulence parameters were all found to change the loads significantly in some simulations, while the mean errors were only slightly affected. This information may, however, be important to include in other situations, e.g. half-wake situations and periods with high shear.
Constraint turbulence simulation was used to constrain the turbulence to match the instantaneously measured wind speeds. Constraining the turbulence to the wind speed measured by the met mast (250 m upstream) increased the errors of the simulated loads. In the simulations, a turbulence event introduced by the constraint turbulence simulator at the met mast is transported unaffected with the steady mean wind to the turbine, in agreement with Taylor's frozen turbulence hypothesis. In the real world, however, the turbulence structures change, and an upstream turbulence event may even pass beside the turbine. The event that



hits the turbine in the simulation is thereby different from the event that hits the real turbine. It is therefore not recommended to use constraint turbulence simulation based on the wind speed measured far away.

Based on pitot-tube wind speed, however, constraint turbulence simulation reduces the mean error of all load sensors in this study. The final case is based on pitot tube mean wind speed, turbulence intensity and shear, and constraint turbulence

simulation based on the pitot-tube-recorded wind speed. In this case, the range of the simulated loads reflects the range of the measured loads. It is therefore assumed to be more suitable for load extrapolation. Moreover, the sequences of the simulated and measured flap and tower-bottom loads are almost similar, meaning that the inflow conditions that result in high load levels in the measurements in most cases also result in high load levels in the simulations and vice versa. The same tendency is seen for the tilt and yaw moment at $14~\mathrm{ms}^{-1}$. In the final case, the range of the simulated loads due to different turbulence

realisations (seeds) decreases considerably, meaning that the need for multiple simulations is reduced.

It was investigated whether the enormous scatter that is seen, especially in the flap and tower-bottom loads, can be predicted by the turbulence intensity, shear profile or atmospheric stability alone. The turbulence intensity explains some of the scatter and the lowest loads are seen in stable conditions with low turbulence intensity and high shear. It is, however, concluded that a more sophisticated approach, which considers the actual combination of inflow parameters, is required to predict the loads of

specific periods

Aeroelastic simulations can be considered to be such an approach. Simulations representing all suitable periods have therefore been performed based on inflow information from the met masts (wind speed, wind-speed trend, turbulence intensity and shear) and the pitot tube (wind speed, wind-speed trend, turbulence intensity, rotor-position-dependent shear and the instantaneously measured wind speed for constraint turbulence simulation). Based on these simulations, it is concluded that HAWC2

simulations based on inflow information from the pitot tube are able to predict the measured flap and tower-bottom load scatter very well in most periods. The met-mast-based simulations yield high loads for most periods in the upper half of the load scatter and vice versa, but the result is less striking.

In both cases, the simulations cannot explain the tilt and yaw moment scatter, as most high-load observations are underestimated at some wind-speed ranges, and low-load observations are overestimated at other wind-speed ranges.

*Data availability.* Simulation results are not available due to confidentiality

*Competing interests.* The authors declare that they have no conflict of interest.

*Acknowledgements.* The authors would like to acknowledge Siemens Wind Power for providing the data for the simulation model, and the funding from the Danish Energy Agency EUDP programme of the DAN-AERO MW projects, contracts ENS no. 33033-0074 and 64009-0258, for providing important data for the present study.





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
