# Peer review of "More accurate aeroelastic wind-turbine load simulations using detailed inflow information"

_Wind Energy Science, 2018_

## Referee Comment (RC1) · Anonymous Referee #1 · 11 Jun 2018

Dear authors and editor,

The paper addresses an interesting aspect of wind turbine development and overall I think is good paper with sufficient details and rationally written.

A minor point is about the references being pretty much focused on work from the authors themselves, while i would suggest to expand (when possible) the list of work done on the same field

The major point is instead that i missed something in the paper. It is very complete, long (33 pages!) and rich of many details but I miss something. I miss the impact of such work on daily life practice in wind turbine development. Using the probe instead of

mast improves the accuracy of the measurements... how much? normally wind turbine manufacturers estimate the impact of new ideas in terms of AEP increase or LCoE reduction. Adding such assessment would increase a lot the value of the paper since is not anymore an academic work but something that helps to bridge the gap between research and industry! It would be for instance good to know what are the costs to equip the turbine with such probe and compare at the end if the costs are worth the improvement!! In terms of blade design, how such improvement could improve the performance of the machine? reducing the severity of certain loads because the design process is more accurate? or perhaps, the life of the component would be extended? by how much? Giving such analysis would make the work significantly more relevant and attractive to read since there is impact on industrial development

best regards

---

## Referee Comment (RC2) · Anonymous Referee #2 · 11 Jun 2018

[referee-annotated manuscript omitted]

---

## Referee Comment (RC3) · Anonymous Referee #3 · 12 Jun 2018

[referee-annotated manuscript omitted]

---

## Author Comment (AC1) · 16 Jul 2018

Thank you very much for a very pleasant review with fine comments and corrections. We have addressed the comments and corrections in the reviewed manuscript.

---

## Author Comment (AC2) · 16 Jul 2018

Thank you very much for a very pleasant review with fine comments and corrections. We have addressed the comments and corrections in the reviewed manuscript.

---

## Author Response (AR1)

**Point by point response to RC1**

1) A minor point is about the references being pretty much focused on work from the authors themselves, while I would suggest to expand (when possible) the list of work done on the same field
   - We will expand the list of references in the reviewed manuscript.
2) The major point is instead that I missed something in the paper.
   It is very complete, long (33 pages!) and rich of many details but I miss something.
   I miss the impact of such work on daily life practice in wind turbine development.
   Using the probe instead of mast improves the accuracy of the measurements. how much?
   - It is not possible to say how much the accuracy of the measurements is improved by using the blade-mounted pitot tube instead of a mast-based flow sensor, as the sensors do not measure the same information. At the current stage, it is therefore more a question about the value of the information that can be measured than the accuracy. A met-mast-based sonic may be highly accurate but it provides only information from a single point, some distance from the turbine while a blade-mounted flow sensor provides information from multiple positions at the rotor plane with some uncertainty, see Pedersen (2018) for more details about the uncertainty.
   In this study the value of the measured information is evaluated in terms of the accuracy of the simulated loads and the use of information from a blade-mounted flow sensor is found to improve the accuracy of these loads.
3) Normally wind turbine manufacturers estimate the impact of new ideas in terms of AEP increase or LCoE reduction. Adding such assessment would increase a lot the value of the paper since is not anymore an academic work but something that helps to bridge the gap between research and industry! […] In terms of blade design, how such improvement could improve the performance of the machine? Reducing the severity of certain loads because the design process is more accurate? or perhaps, the life of the component would be extended? by how much?
   - The demonstrated application of blade-mounted flow sensors does not improve AEP, LCoE or the blade design. The instrument is, however, able to characterize some of the conditions that results in high loads and may help to predict future loads with higher accuracy. In addition, the demonstrated one-to-one approach may decrease the measurement period that is required for load validation as the aeroelastic model can be validated based on single time series instead of the average of a large measurement dataset. At the current stage, it is therefore only a research instrument, and only time can tell the industrial value. Measurements from blade-mounted flow sensors may, however, potentially be used as input to control of cyclic or individual pitch or active trailing edge flaps to increase power or reduce loads and noise (Larsen et al., 2005; Barlas et al., 2012; Kragh and Hansen, 2012; Kragh et al., 2012; Madsen, 2014). The expected load reduction and component life-time extension potential is highly dependent on the strategy and the actuator technology and specifications.
4) It would be for instance good to know what are the costs to equip the turbine with such probe and compare at the end if the costs are worth the improvement!!

- To our knowledge no wind-turbine or pitot-tube manufactures offers a ready-to-use pitot-tube sensor system for wind turbines. Swiss Air-Data (www.swiss-airdata.com) sells a five-hole air data boom that provides calibrated and aerodynamically corrected air data for 11,200 Euro. This system, however, does not include data acquisition, power supply or blade-mounting solutions.
* * *
Barlas, T. K., van der Veen, G. J. and van Kuik, G. A. M.: Model predictive control for wind turbines with distributed active flaps: incorporating inflow signals and actuator constraints, Wind Energy, 15(5), 757–771, doi:10.1002/we.503, 2012.

Kragh, K. and Hansen, M.: Individual Pitch Control Based on Local and Upstream Inflow Measurements, in 50th AIAA Aerospace Sciences Meeting including the New Horizons Forum and Aerospace Exposition 09-12 January 2012, Nashville, Tennessee, American Institute of Aeronautics and Astronautics., 2012.

Kragh, K. A., Henriksen, L. C. and Hansen, M. H.: On the Potential of Pitch Control for Increased Power Capture and Load Alleviation, in Torque, the science of making torque from wind., Presented at The Science of Making Torque from Wind 2012, 9-11 October 2012, Oldenburg, Germany. [online] Available from: www.orbit.dtu.dk, 2012.

Larsen, T. J., Aagaard Madsen, H. and Thomsen, K.: Active load reduction using individual pitch, based on local blade flow measurements, Wind Energy, 8(1), 67–80, doi:10.1002/we.141, 2005.

Madsen, H. A.: Correlation of amplitude modulation to inflow characteristics, Proc. 43rd Int. Congr. Noise Control Eng. [online] Available from: http://www.acoustics.asn.au/conference_proceedings/INTERNOISE2014/papers/p171.pdf, 2014.

Pedersen, M. M., Larsen, T. J., Madsen, H. A. and Andersen, S. J.: Free-flow wind speed from a blade-mounted flow sensor, Wind Energy Sci., 3(1), 121–138, doi:10.5194/wes-3-121-2018, 2018.

**Point by point response to RC2**

1) Excellent paper, some small suggestions and comments in the attached pdf.
   - Thank you very much for a very pleasant review with fine comments and corrections. We have addressed the comments and corrections in the reviewed manuscript.

**Point by point response to RC3**

1. It is an excellent paper in terms of scientific significance. Some comments can be found in the attached "pdf" file.
   - Thank you very much for a very pleasant review with fine comments and corrections. We have addressed the comments and corrections in the reviewed manuscript.

**Points from the suplement**

2. It would be more clear if you can describe these moments using a coordinate system. Are these moment sensors measured the moment in "not-pitched" reference frame or in "pitched" reference frame?
   - An illustration (Fig. 4) showing the orientation has been inserted
3. What is this "another approach"? Could you please explain it a bit more?
   - It is a wind shear model, e.g. the power shear model. The sentence has been revised.
4. "mean absolute relative error" Is this absolute error or relative error?
   - It is the absolute value of the error relative to the measured load. I have replaced the phrase with "mean relative error"
5. The difference between Siemens controller and the Basic controller can in general affect other test cases. Why they are not mentioned in the cases from "Case 1" to "Case 4"?
   - It is now mentioned in a general comment to the simulation model.

[revised manuscript text omitted]

---

## Author Response (AR2)

Thank you for your comments and suggestions for a more decent structure.

We have followed your advices and shortened the introduction considerable. In return, we have added a new section - "Problem discussion" - with some of the contents from the previous introduction and removed the paragraph that mentions the possibility of using lidars.

5    We have moreover split the previous method section into a "Measurements" and a "Simulations" section.

The paragraph that mentions effects of rotational sampling is revised, to avoid that the reader gets the impression that we have discovered these effects.

We have not elaborated the discussion of rotational sampling as we are not considering or using the rotational sampled turbulence characteristics (except for the frame-of-reference-independent uu-spectra integral, which is used to determine the

10   $\alpha\epsilon^{2/3}$ Mann parameter).

We have finally included the values found in the references to quantify the uncertainties of the wind speed estimated from blade mounted flow sensors and several paragraphs are revised to avoid misconception

[revised manuscript text omitted]

---

## Author Response (AR3)

Changes prompted by comments from chief editor:
- "met mast" changed to "meteorological mast" in the abstract
- "met mast" defined the first time it is used after the abstract.

[revised manuscript text omitted]